# Novel Iron Oxide Nanoparticles Induce Ferroptosis in a Panel of Cancer Cell Lines

**DOI:** 10.3390/molecules27133970

**Published:** 2022-06-21

**Authors:** Roberto Fernández-Acosta, Claudia Iriarte-Mesa, Daniel Alvarez-Alminaque, Behrouz Hassannia, Bartosz Wiernicki, Alicia M. Díaz-García, Peter Vandenabeele, Tom Vanden Berghe, Gilberto L. Pardo Andreu

**Affiliations:** 1Department of Pharmacy, Institute of Pharmaceutical and Food Sciences, University of Havana, 222 Street # 2317, La Coronela, La Lisa, Havana 13600, Cuba; roberto.fernandezac91@gmail.com; 2Laboratory of Bioinorganic (LBI), Department of Inorganic and General Chemistry, Faculty of Chemistry, University of Havana, Zapata y G, Vedado, Plaza de la Revolución, Havana 10400, Cuba; iriarte190793@gmail.com (C.I.-M.); adg@fq.uh.cu (A.M.D.-G.); 3Institute of Inorganic Chemistry—Functional Materials, University of Vienna, Währinger Straße 42, 1090 Vienna, Austria; 4Center for Research and Biological Evaluations, Institute of Pharmaceutical and Food Sciences, University of Havana, 222 Street # 2317, La Coronela, La Lisa, Havana 13600, Cuba; dalminaque.85@gmail.com; 5VIB Center for Inflammation Research (IRC), 9052 Ghent, Belgium; behrouz.hassannia@irc.vib-ugent.be (B.H.); bartoszw@irc.vib-ugent.be (B.W.); peter.vandenabeele@irc.vib-ugent.be (P.V.); tom.vandenberghe@uantwerpen.be (T.V.B.); 6Department of Biomedical Molecular Biology (DBMB), Ghent University, 9052 Ghent, Belgium; 7Methusalem Program, Ghent University, 9052 Ghent, Belgium; 8Laboratory of Pathophysiology, Department of Biomedical Sciences, University of Antwerp, 2000 Antwerp, Belgium; 9Ferroptosis and Inflammation Research (FAIR), VIB Research Center, Ghent University, 9052 Ghent, Belgium; 10Ferroptosis and Inflammation Research (FAIR), University of Antwerp, 2000 Antwerp, Belgium

**Keywords:** IONP–GA/PAA, ferroptosis, cancer cells

## Abstract

The use of nanomaterials rationally engineered to treat cancer is a burgeoning field that has reported great medical achievements. Iron-based polymeric nano-formulations with precisely tuned physicochemical properties are an expanding and versatile therapeutic strategy for tumor treatment. Recently, a peculiar type of regulated necrosis named ferroptosis has gained increased attention as a target for cancer therapy. Here, we show for the first time that novel iron oxide nanoparticles coated with gallic acid and polyacrylic acid (IONP–GA/PAA) possess intrinsic cytotoxic activity on various cancer cell lines. Indeed, IONP–GA/PAA treatment efficiently induces ferroptosis in glioblastoma, neuroblastoma, and fibrosarcoma cells. IONP–GA/PAA-induced ferroptosis was blocked by the canonical ferroptosis inhibitors, including deferoxamine and ciclopirox olamine (iron chelators), and ferrostatin-1, the lipophilic radical trap. These ferroptosis inhibitors also prevented the lipid hydroperoxide generation promoted by the nanoparticles. Altogether, we report on novel ferroptosis-inducing iron encapsulated nanoparticles with potent anti-cancer properties, which has promising potential for further in vivo validation.

## 1. Introduction

The link between nanoparticles and medicine can be traced to the 1970s, although the era of nanomedicine started at the turn of this century, with more than 1000 publications a year since 2010 [1]. So far, about 50 nanodrugs are currently available for clinical use, from which the anticancer polymeric nano-formulations are the most common [2]. Cancer is one of the most challenging conditions, and there is increasing interest in developing advanced nanostructured formulations for its therapy. Particularly, iron-containing nanomaterials, like iron oxide nanoparticles (IONP), have been identified as potent tools for anticancer research, including targeted drug delivery, bioimaging, biosensors, hyperthermia, and selective cancer cell death induction [3,4,5,6,7,8,9]. IONP can be classified based on the different oxidation states and crystalline structures, such as magnetite (Fe_3_O_4_), maghemite (*γ*–Fe_2_O_3_), and hematite (*α*–Fe_2_O_3_). Such materials have shown high biocompatibility and therapeutic efficacy as was proven by Nanotherm™ (amino silane-coated IONP designed for glioblastoma therapy using local tissue hyperthermia), which features among the FDA-approved iron oxide-based nanomedicines [10,11].

On the other hand, the synthesis of nanomaterials with intrinsic anticancer activity as activators of regulated cell death has lately attracted extensive attention [12]. Beyond the induction of apoptosis, recent studies identified several nanomaterials inducing ferroptosis as an anticancer strategy [4,12,13,14,15,16,17]. Ferroptosis is a clinically relevant distinct mode of regulated necrosis recently described as an iron-catalyzed process of excessive lipid peroxidation [18,19,20,21]. It was suggested that the observed anticancer activity of some iron-based nanoparticles are related to ferroptosis induction as a result of the release of ferrous or ferric ions in acidic lysosomes after endocytosis leading to Fenton reactions, which, in turn, can produce ROS, lipid peroxidation, intracellular macromolecules damage, and ferroptosis [12,21,22].

Considering the several limitations of conventional therapies against cancer, the search for tumor-specific targeting systems which maximize the therapy efficiency with minimal side effects is a challenging issue in current cancer therapy [23,24]. The employment of nanoparticles is a promising alternative due to, for example, that tumor vasculature enhances the accumulation and retention of these systems, a phenomenon known as the enhanced permeability and retention (EPR) effect [24].

We evaluated the anticancer activity of novel magnetic IONP functionalized with gallic acid (GA) and polyacrylic acid (PAA). These nanoparticles (IONP–GA/PAA) were previously used as support for the immobilization of *Trametes versicolor* laccase and subsequent removal of organic dyes [25]. Given their enhanced colloidal stability conferred by the organic coatings and interesting magnetic properties, we proceeded to evaluate the potential biomedical applications of this material (Figure 1). Here, we show for the first time that the referred nanosystem possesses precisely tuned physicochemical properties for the induction of intrinsic anticancer activity mediated by ferroptosis execution.

## 2. Results

### 2.1. Synthesis and Characterization of IONP–GA/PAA

IONP–GA/PAA were obtained according to a previously reported procedure [25] based on the thermal decomposition of an iron (III) oleate complex (IONP–OA/TOPO) followed by the ligand exchange with GA and PAA (IONP–GA/PAA) (Figure 2A). IONP–OA/TOPO exhibited spherical morphology and a size of 13 nm (Figure 2B), which was preserved after the ligand exchange, as can be seen in Figure 2C. The hydrophobic coating of OA and TOPO (6% *w*/*w*) was totally replaced after the ligands exchange, which enhanced colloidal stability in water due to the presence of GA and PAA moieties (57% *w*/*w*). The IONP–GA/PAA presented a hydrodynamic diameter of 70 ± 20 nm, which did not change for at least 4 months after ligand exchange (Figure 2D). The value of ζ-potential obtained for the IONP–GA/PAA at pH 7 (−48 mV) confirmed the high negative surface charge of the colloidal system, mainly due to the presence of the PAA polymer grafted (Figure 2E).

The magnetic properties obtained through Vibrating Sample Magnetometry (VSM) are summarized in Table 1. The nanoparticles exhibited superparamagnetic behavior with insignificant values of intrinsic coercivity (H_c_). The decrease in saturation magnetization (σ_s_) after the ligand exchange was attributed to the high amount of organic material grafted on the surface of the IONP–GA/PAA. The characterization of the nanoparticles by X-ray powder diffractometry (XRD) and Fourier-transform infrared spectroscopy (FTIR) corresponded with the results previously reported [25] and confirmed the obtaining of magnetite (Fe_3_O_4_) as major phase after the ligand exchange, as well as the successful replacement of hydrophobic coatings of TOPO and OA by GA and PAA. The characterization results validated the reproducibility of the method chosen for the synthesis of the IONP–GA/PAA.

### 2.2. IONP–GA/PAA Induce Cell Death in a Panel of Cancer and Non-Tumorigenic Cell Lines

We examined the sensitivity of a panel of cancer cell lines to IONP–GA/PAA. IONP–GA/PAA significantly killed glioblastoma (U87MG and U373MG), high-risk MYCN amplified neuroblastoma (IMR32), fibrosarcoma (HT1080), and HT22 neuronal (non-tumorigenic) cell lines (Figure 3A). The HT1080 cells reached 100% cell death in 24 h (Figure 3A,C), while the other cell lines only reached 100% cell death at 48 h (Figure 3A). As a control, no cell death was observed by using only gallic acid (GA) at 6 µg/mL (a similar concentration to that carried on the IONP–GA/PAA surface), discarding the possibility of a gallic acid-mediated cytotoxic effect.

Figure 3B shows the dose-dependent effect of IONP–GA/PAA on HT1080 in 24 h. The EC_50_ (half-maximal effective concentration) was 2.97 µg/mL. Figure 3D shows the kinetic profile of the cytotoxic effects of several concentrations of IONP–GA/PAA on HT1080 cells with a positive correlation between the concentration and the slopes.

### 2.3. IONP–GA/PAA Induced-Cell Death in HT1080 Cells Is Blocked by Ferroptosis and Heme Oxygenase-1 Inhibitors

HT1080 cells were pre-treated with a panel of apoptotic and non-apoptotic cell death inhibitors before exposing them to IONP–GA/PAA. IONP–GA/PAA-induced cell death was blocked by a panel of canonical ferroptosis inhibitors: the lipophilic free radical trap ferrostatin-1 (Fer1), and the iron chelators deferoxamine (DFO) and ciclopirox olamine (CPX) (Figure 4A). The pan-caspase inhibitor, Z-VAD-FMK, and the necroptosis inhibitor, necrostatin-1 (Nec-1s, a RIPK1 kinase inhibitor), did not block the cell death, nor did the fluorescent dye DEVD-AMC show caspase-3 activity (Figure 4A).

Finally, zinc protoporphyrin (ZnPP) also blocked the IONP–GA/PAA-triggered cell death by 55% (Figure 4B), suggesting the involvement of heme oxygenase-1 (HMOX1) in the mechanism of cell death induced by the nanoparticles, as previously reported [26,27].

### 2.4. IONP–GA/PAA Induced-Lipid Peroxidation in HT1080 Cells Is Blocked by Ferroptosis Inhibitors

Considering that lipid peroxidation drives ferroptotic cell death, we analyzed this parameter after IONP–GA/PAA exposition by using the lipophilic and oxidation-sensitive fluorochrome C11-BODIPY. We found that IONP–GA/PAA induced an early wave of lipid peroxidation in HT1080 fibrosarcoma cells with a four-fold increase at 30 min of treatment (Figure 5A), which then begins to decrease at 2 h and 6 h of treatment when plasma membrane permeabilization starts to occur (Figure 5). This early wave of lipid peroxidation prior to membrane permeabilization is a typical pattern of ferroptotic cell death [28]. Iron chelation using DFO blocked both IONP–GA/PAA-triggered lipid peroxidation and cell death induction, as measured by DRAQ7 uptake (Figure 5). Likewise, we observed the same complete protection against lipid hydroperoxide generation and cell death by using Fer1 (Figure 5).

## 3. Discussion

In this study, we present pieces of evidence suggesting the induction of ferroptosis in cancer cells by novel iron oxide nanoparticles. After immobilizing laccase enzymatic activity on these IONP–GA/PAA, which demonstrated the proof of principle of using them as a biocatalytic nanosystem for industrial applications [25]. This is the first report that points out the potentiality of this nanomaterial to be employed as therapeutics for tumor treatment. The composition and physicochemical characteristics of the IONP–GA/PAA confirm their rational engineering as an anticancer nanomedicine.

Polyacrylic acid is a weak polyelectrolyte that can be used to enhance the colloidal stability of magnetite nanoparticles (or other nanosystems) in an aqueous medium [29,30]. It also confers biocompatibility and versatility for biomedical applications due to the capacity of its carboxylate moieties (see Figure 1), which can be functionalized with several anti-neoplastic agents like bleomycin and doxorubicin, while retaining the important superparamagnetic behavior of the magnetic core [31,32,33,34,35]. Additionally, PAA endows magnetite with low plasma protein adsorption which improves the blood circulation, thus favoring the nanocomposite magnetite-PAA to be used for drug delivery. It also enhances the stability of other coatings agents like chitosan, which is important for drug-controlled release [35,36]. Likewise, PAA-coated magnetite was identified as a good pH-responsive nanomaterial because of the different forms (polyacrylic acid and polyacrylate) of PAA under variable pH solvents, which tunes the coating polymer’s surface solubility. This, in turn, controls the binding force between the PAA and the drug and regulates the delivery performance [37]. On the other hand, gallic acid-functionalized magnetite was reported as a highly advantageous system in biomedicine, with special attention to theragnostic: magnetic resonance imaging and magnetic hyperthermia, and also as a nanocarrier of GA for antioxidant, antimicrobial, and anticancer effects [38,39,40,41]. Combining important properties of GA and PAA, like the stabilization of the aqueous magnetic fluid, biocompatibility, and avoiding interactions with whole blood, the GA/PAA-coated magnetite nanocomposite constitutes a real asset for cancer nanomedicine.

The novel IONP–GA/PAA studied here consists of magnetite as the predominant phase (however, a mixture of magnetite/maghemite cannot be ruled out), in which about 57% of the total weight corresponds to the organic coating (GA and PAA), and the ratio of GA/PAA was estimated to be roughly 9:1 [25]. Both polyacrylic and gallic acid render iron oxide nanoparticles water-soluble, and the presence of gallic acid hinders the formation of multiple polyacrylic acid layers, therefore improving the colloidal stability of the nanoparticles, ensuring a high batch-to-batch reproducibility, and achieving stability in their physicochemical properties during at least 4 months after the ligand exchange reaction [25]. The IONP–GA/PAA are nearly monodisperse sphere-shaped nanoparticles in the multi-gram scale with an average core diameter of 13 nm and hydrodynamic diameter of around 70 nm in water at pH 7, also exhibiting a superparamagnetic behavior that is retained at room temperature [25]. Such physicochemical properties also meet the general specifications for biomedical and bioengineering applications, which require that these nanoparticles have high magnetization values, sizes smaller than 100 nm, and an overall narrow particle size distribution such that the particles have uniform physical and chemical properties, batch-to-batch reproducibility, and stability in time (for storage) [42,43,44,45]. The superparamagnetic behavior confers versatility to the system due to the possible hyperthermia induction and the use of the magnetic field to concentrate these iron-based nanoparticles at the tumor site [22].

Once we examined the potentialities of the referred IONP–GA/PAA system for biomedicine and being inspired by its recently shown industrial and environmental applications [25], we decided to start evaluating its anticancer effect. IONP–GA/PAA induced 100% of cell death in a panel of human cancer cell lines, including HT1080 fibrosarcoma, U87MG and U373MG glioblastoma, and IMR32 neuroblastoma. A cytotoxic effect was also found in a non-tumorigenic cell line (HT22) at 48 h. It is important to emphasize that even when IONP–GA/PAA exert a significant toxic effect on a normal cell line, its selectivity can only be approached in in vivo assays. In this regard, it has been shown that, in general, two molecular effects make cancer cells more vulnerable to ferroptosis induction by IONP: enhanced permeability and retention effect and iron “addiction”. The first is related to the presence in the tumorous tissue of leaky vessels and pores which enhance the accumulation of nanoparticles at the tumor site [24], and the second describes the higher iron requirements of cancer cells in comparison with normal cells [46,47]. Both of these molecular phenomena may be exploited for therapeutic benefits and should be approached in future experimental designs with this IONP.

Notoriously, the IONP–GA/PAA concentration used here was 3.3 µg/mL, which is below the threshold concentration of superparamagnetic iron oxide nanoparticles to avoid oxidative stress-induced cell injury and death in normal cell lines [48]. Even better, according to the kinetic profile, good values can be achieved with a nanoparticle’s concentration three times lower (1.1 µg/mL). The kinetic profile also shows an increase in the curve slope after 16 h of treatment, which suggests an enhancement in the cell death induction, probably mediated by a higher rate of incorporation of the nanoparticles into the cells.

IONP–GA/PAA-induced cell death in fibrosarcoma cells lacks caspase-3 activation (a characteristic step in apoptosis execution), and it is not affected by necroptosis (Nec-1s) or pan-caspase inhibitors (zVAD-fmk). Conversely, the cell death was blocked by the canonical ferroptosis inhibitors, ferrostatin-1 (small-molecule radical-trapping agent), deferoxamine, and ciclopirox [49,50]. The last two referred inhibitors are iron-chelating agents. Nanoparticles enter cells by different endocytic pathways and are transported into endosomes via endosomal-lysosomal pathway. In this low pH condition, degradation of the nanoparticles occurs with subsequent iron ions release [51], leading to an increase in the iron labile pool and lipid peroxidation accumulation via Fenton reaction. In this context, iron chelators like DFO and CPX can prevent iron-dependent lipid ROS production and ferroptosis by sequestering iron ions. Elucidation of specific mechanisms for iron release from these nanoparticles are still to be approached by our group. Likewise, the same inhibitors (Fer1 and DFO) blocked the IONP–GA/PAA-mediated increase of lipid peroxidation exerted by the nanoparticles, which highlights the catalytic role of iron in the oxidation of the polyunsaturated fatty acid of the cellular membrane.

Finally, we suggest that IONP–GA/PAA-induced cell death involves the activation of heme oxygenase-1. HMOX1 activity is a major intracellular source of free iron which has been identified as an essential enzyme for iron-dependent lipid peroxidation during ferroptotic cell death [27,52,53]. Enforced by excessive cellular iron and ROS, this enzyme changes from a cytoprotective role against cell death (by scavenging ROS) to one of a perpetrator (by increasing labile Fe^2+^ pool, leading to ROS overload and ferroptosis induction) [26,52,53]. In line with this, we observed that the competitive HOMX1 inhibitor zinc protoporphyrin [54] prevented more than 50% of the IONP–GA/PAA-triggered cancer cell death, which points out the HOMX1 implication in its mechanism for ferroptosis execution.

All the aforementioned results allow us to conclude that the IONP–GA/PAA nanoparticles induce a cell death mechanism that presents several hallmarks of ferroptosis. Even though more experimentation is required to establish the precise mechanism for the ferroptosis induction, and to define better applications of these nanoparticles for cancer therapy, this work expands the current panorama of iron oxide nanoparticles and ferroptosis-based cancer therapy. To the best of our knowledge, this is the first work reporting ferroptosis induction by the sole effect of magnetite-based nanomaterials. On the other hand, a recently published work evaluated the ferroptosis induced by another system also based on Fe_3_O_4_ nanoparticles in combination with gene interference (DMP controlled CRISPR/Cas13a knockdown of two iron metabolic genes, FPN and LCN2) for cancer therapy [55], which confirms the versatility and multiple therapeutic strategies that can be followed by using rationally designed iron oxide nanoparticles.

## 4. Materials and Methods

### 4.1. Reagents

All reagents and solvents employed for the synthesis of the IONP were commercially available high-grade purity (Aldrich Chemicals) and were used as supplied without further purification. In addition, the following reagents were used: BODIPY 581/591 C11 probe (Invitrogen, Waltham, MA, USA, D-3861) was used at 2 μM, DRAQ7 (BioStatus, Loughborough, UK, DR71000) was used at 0.3 μM, SytoxGreen (Thermo Fisher Scientific, Waltham, MA, USA, S7020) was used at 1.7 μM, DEVD-AMC (Pepta Nova, Sandhausen, Germany, 3171-V) was used at 20 μM, Nec-1s (Calbiochem, San Diego, CA, USA, 480065) was used at 10 μM, Z-VAD-FMK (Bachem, Bubendorf, Switzerland, N-1510) was used at 10 μM, Fer1 (Xcess Biosciences, Harlem & Touhy Plaza, Chicago, IL, USA, 053224) was used at 1 µM, DFO (Sigma-Aldrich, St. Louis, MO, USA, D-9533) was used at 50 μM, CPX (Sigma-Aldrich, St. Louis, MO, USA, C0415) was used at 5 μM, and ZnPP (Enzo Life Sciences, Farmingdale, New York, NY, USA, ALX-430-049-M025) was used at 1 μM.

### 4.2. Synthesis of the Iron Oxide Nanoparticles

The nanoparticles were obtained according to a previously described methodology [25]. In brief, magnetite nanoparticles were synthesized by thermal decomposition and later transferred to the water by a ligand exchange method with polyacrylic acid and a polyacrylic acid/gallic acid mixture.

In a 50 mL round-bottom flask, iron (III) oleate (800 mg) was mixed with OA (70 µL) and TOPO (61.7 mg) and dissolved in 3 mL of dioctyl ether (DOE) at 100 °C. The mixture was kept at 100 °C for 1 h under N_2_ atmosphere and then the temperature was increased to 310 °C for another 1.5 h. The nanoparticles were precipitated by centrifugation (10,000 rpm) and washed several times with acetone and diethyl ether. The well-dried powder was redispersed in chloroform (5 mg/mL) for the subsequent ligand exchange process.

The stock solution of IONP–OA/TOPO (1 mL) was mixed with 30 mL of water solution of PAA (100 mg, 100 kDa) and GA (50 mg). The biphasic chloroform/water system was kept under sonication during 20 min and subsequent stirring for 24 h. After the nanoparticles were transferred to water, EtOH (20 mL) and n-hexane (10 mL) were added, and the mixture was centrifuged (7000 rpm). The nanoparticles were washed several times with an EtOH/water mixture and the well-dried powder was redisposed in water at pH 7 (5 mg·mL^−1^).

### 4.3. Evaluation of Physicochemical Properties of the Iron Oxide Nanoparticles

Transmission electron microscopy (TEM) images were obtained from a Philips CM30 microscope with an accelerating voltage of 300 kV. The analysis of the TEM images was performed with Image J. The hydrodynamic size of the IONP–GA/PAA in aqueous solution was evaluated using a StabiSizer PMX 200C from Particlemetrix. The evaluation of ζ-potential was carried out at 25 °C by electrophoretic light scattering (DLS) using the Anton Paar Litesizer TM 500. Infrared spectra were recorded in a spectrometer, WQF-510 FTIR from Rayleigh, using tablets of potassium bromide to prepare the samples. X-ray powder diffraction patterns were obtained in an X’pert-Pro powder diffractometer from PANalytical (Cu Kα with λ = 1.54 Å). The magnetic properties of the samples were determined using in an Evercool Physical Properties Measurements System (PPMS P525 Quantum Design) in the Vibrating Sample Magnetometry (VSM) mode. The hysteresis loop of the samples was registered at 300 K with a maximum field of 70 kOe and a sensibility of 10^−6^ emu.

### 4.4. Cell Culture Conditions

HT22, U87MG, and U373MG cells were cultured in DMEM medium supplemented with 10% FCS (*v*/*v*), l-glutamine (1 mM), sodium pyruvate (1 mM), and nonessential amino acids (1 mM). IMR-32 cells were cultured in RPMI 1640 medium supplemented with 10% FCS (*v*/*v*), l-131 glutamine (1 mM), sodium pyruvate (1 mM), and non-essential amino acids (1 mM). HT1080 cells were cultured in EMEM medium supplemented with 10% FCS (*v*/*v*), l-glutamine (1 mM), sodium pyruvate (1 mM), and non-essential amino acids (1 mM). IMR-32 cells were obtained from Jo Vandesompele, Ghent University Hospital, Medical Research Building, Ghent, Belgium. HT22, U87MG, U373MG, and HT1080 cells were obtained from ATCC. Cells were cultured at 37 °C in a humidified atmosphere containing 5% CO_2_ and split every 3–4 days using trypsin/EDTA solution.

### 4.5. Analysis of Cell Death and Caspase-3 Activity

Cell death and caspase-3 activity were measured as previously described, using the FLUOstar Omega fluorescence plate reader (BMG Labtech GmbH) [56]. This fluorescent apoptosis/necrosis assay monitors cell death as an increase in fluorescence intensity of a cell-impermeable dye (SytoxGreen) after plasma membrane disintegration, whereas apoptosis is detected through caspase-mediated release of a fluorophore from its quencher (DEVD-AMC) [56]. In short, cells were seeded in a 96-well plate and all experiments were performed at least in triplicate. The next day, cells were preincubated with the desired inhibitors for 1 h or 4 h (DFO), and then treated with stimuli at desired concentrations in the presence of SytoxGreen (1.7 μM) and DEVD-AMC (20 μM). After that, the plate was transferred to a temperature- and CO_2_-controlled FLUOstar Omega fluorescence plate reader. The fluorescence intensity of SytoxGreen and DEVD-AMC was measured in function of the time at intervals of 1 h with excitation/emission filters of 485/520 nm for SytoxGreen and 360/460 nm for DEVD-AMC. In each experiment, Triton X-100 (0.05%) was used to induce lysis of the cells in at least 3 wells of the plate, and its signal intensity was used as a 100% cell death reference. The percentage of the cell death was calculated by the following formula: (avg. SytoxGreen [stimuli] − avg. SytoxGreen [background])/(avg. SytoxGreen [Triton X-100] − avg. SytoxGreen [background]) × 100. Caspase-3 activity was calculated by subtracting the fluorescence intensity of DEVD-AMC of the treated cells from the control (untreated cells). Cell death was also analyzed by DRAQ7 (0.3 μM) staining coupled with flow cytometry using BD LSRFortessa (BD Biosciences, San Jose, CA, USA).

### 4.6. Live-Cell Imaging

Live-cell images of cells seeded in a 96-well plate (following the same methodology used to analyze cell death) were acquired with an LSM780 confocal microscope (Zeiss, Jena, Germany). Image merging was performed on ImageJ software.

### 4.7. Lipid ROS Assay

HT1080 (300,000 cells/well) cells were seeded in a 6-well plate. The next day, cells were treated with stimuli and harvested. At 10 min before each time point, fluorescent probes were added to proper wells: 2 μM C11-BODIPY and 0.3 μM of DRAQ7. Lipid ROS accumulation was measured at specific time points using BD LSRFortessa (BD Biosciences). Fluorescence was measured in B530 (C11-BODIPY) and R780 (DRAQ7) channels. Only fluorescence of not permeabilized cells (live cells) was analyzed. A minimum of 10,000 cells were analyzed per condition. C11-BODIPY changed its fluorescence properties upon oxidation. Specifically, oxidation of the polyunsaturated butadienyl portion of C11-BODIPY resulted in a shift of the fluorescence emission peak from ≈590 nm to ≈510 nm. On the other hand, C11-BODIPY resides in lipophilic membrane structures where it can be oxidized by different kinds of radicals. Its oxidation is therefore an indirect manifestation of the lipid ROS increase [28,57].

### 4.8. Statistical Analysis

All statistical analyses were performed using GraphPad Prism 9.2.0 software. Data are presented as mean ± SD from three independent experiments. *p* values were calculated with Student’s unpaired *t* test (* *p* < 0.05, ** *p* < 0.01, *** *p* < 0.001, **** *p* < 0.0001; see figure legend for further detail).

## 5. Conclusions and Perspectives

Here, we show for the first time that novel iron oxide nanoparticles, IONP–GA/PAA, possess intrinsic anticancer activity by triggering ferroptosis in fibrosarcoma cells. Remarkably, this polymeric nano-formulation turned out to be an asset for industrial application, which speaks loudly about the adaptability of this multitasking nanomaterial, thanks to its composition, reproducibility, and physicochemical properties. This work opens the possibility to study the use of IONP–GA/PAA for biomedical research and application in the field of cancer nanomedicine.

## Figures and Tables

**Figure 1 molecules-27-03970-f001:**
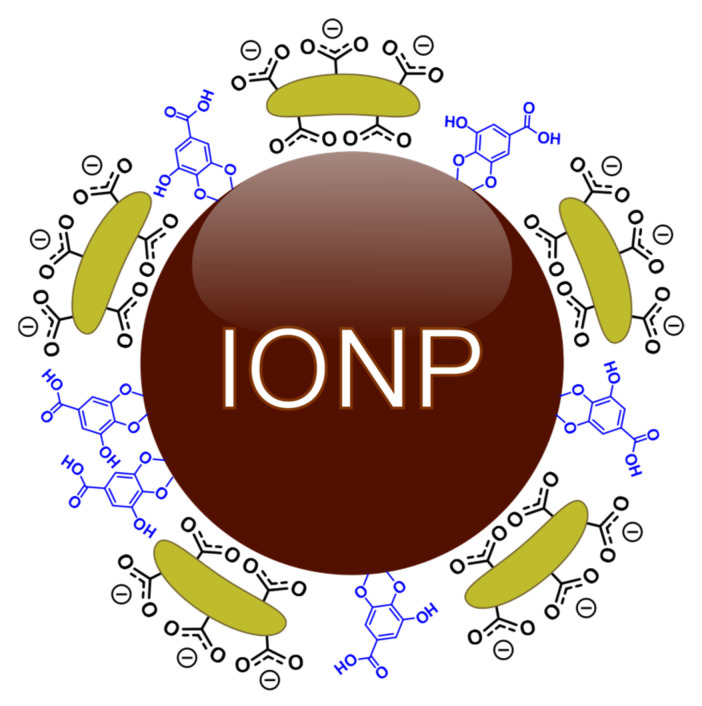
Ligand coverage of polyacrylic acid (green) and gallic acid (blue) coordinate to the surface of magnetite (Fe_3_O_4_) to form the Fe_3_O_4_-GA/PAA nanocomposite.

**Figure 2 molecules-27-03970-f002:**
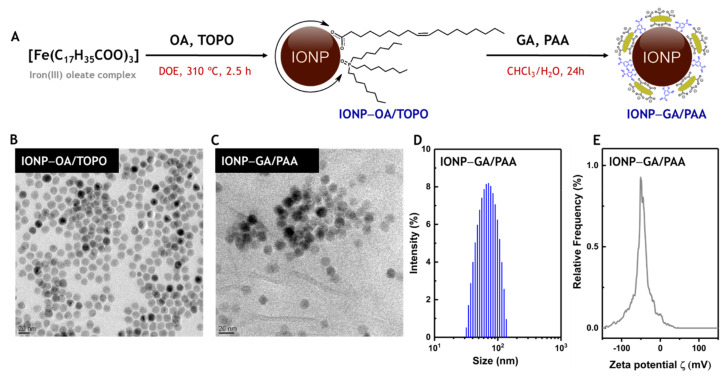
Synthesis and characterization of iron oxide nanoparticles functionalized with gallic acid and polyacrylic acid (IONP–GA/PAA). (**A**) Synthesis of IONP–GA/PAA by thermal decomposition and subsequent ligand exchange. (**B**,**C**) Transmission electron microscopy (TEM) images of the IONP–OA/TOPO and IONP–GA/PAA, respectively. Scale bars correspond to 20 nm. (**D**) Dynamic light scattering (DLS) profile of the IONP–GA/PAA. (**E**) ζ-potential profile of the IONP–GA/PAA.

**Figure 3 molecules-27-03970-f003:**
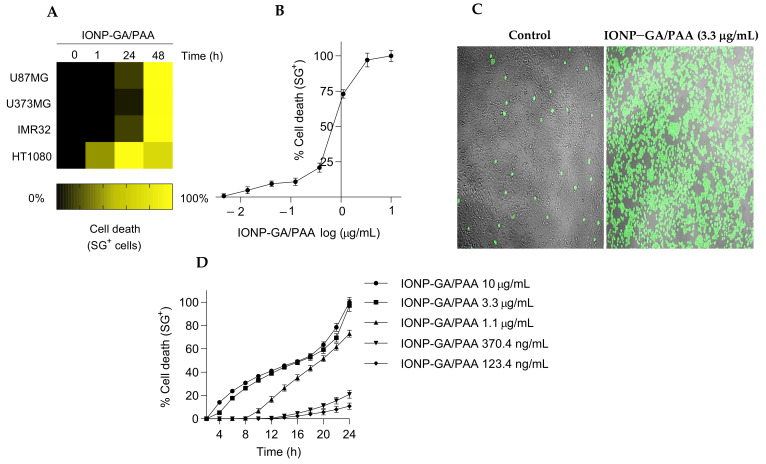
IONP–GA/PAA induce cell death in a panel of cancer and non-tumorigenic cell lines. (**A**) Heatmap representing cell death sensitivity of various cancer cell lines after exposure to IONP–GA/PAA (3.3 μg/mL): HT22 (mouse hippocampal neuronal cell line), U87MG and U373MG (human glioblastoma cell lines), IMR32 (human neuroblastoma cell line), and HT1080 (human fibrosarcoma cell line). Each cell line was incubated with IONP–GA/PAA for 48 h. (**B**) Dose-effect curve of the effect of IONP–GA/PAA on HT1080 cell line during 24 h. (**C**) Snapshots from live-cell imaging of untreated (control) and IONP–GA/PAA-treated cells (3.3 µg/mL, 24 h). Green fluorescent staining represents SytoxGreen dye. (**D**) Kinetic profile of the effect of several concentrations of IONP–GA/PAA on HT1080 cells. Measurements were taken every 2 h up to 24 h.

**Figure 4 molecules-27-03970-f004:**
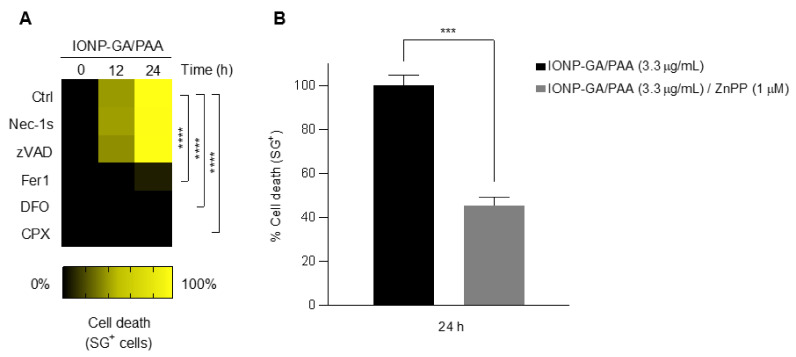
IONP–GA/PAA-induced cell death is blocked by ferroptosis and heme oxygenase-1 inhibitors. (**A**) Heatmap representing cell death sensitivity of HT1080 cells after exposure to 3.3 µg/mL of IONP–GA/PAA, in the absence or presence of different inhibitors. Necroptosis inhibitor: necrostatin-1 (Nec-1s, 10 µM). Pan-caspase inhibitor: Z-VAD-FMK, 10 µM. Ferroptosis inhibitors: the lipophilic free radical trap and lipid peroxidation inhibitor, ferrostatin-1 (Fer1, 1 µM); and the iron chelators, deferoxamine (DFO, 50 µM) and ciclopirox olamine (CPX, 5 µM). (**B**) Percentage of cell death induced by IONP–GA/PAA (3.3 µg/mL, 24 h) in the presence/absence of HMOX1 inhibitor zinc protoporphyrin (ZnPP, 1 µM). Data are presented as mean ± SD from three independent experiments. *p* values were calculated with Student’s unpaired *t* test (*** *p* < 0.001, **** *p* < 0.0001).

**Figure 5 molecules-27-03970-f005:**
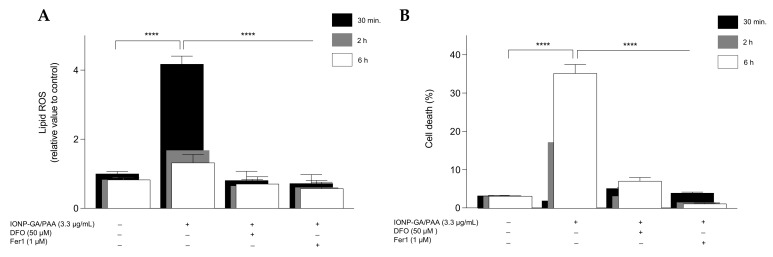
IONP–GA/PAA induced-lipid peroxidation in HT1080 cells is blocked by ferroptosis inhibitors. (**A**) Flow cytometry analysis of the lipid peroxidation sensor (C11-BODIPY-581/591 dye) on live-gated cells (DRAQ7-negative cells) after treatment of HT1080 cells with IONP–GA/PAA (3.3 µg/mL) for 30 min, 2 h, and 6 h in absence or presence of the following ferroptosis inhibitors: DFO (50 µM) and Fer1 (1 µM). (Values are expressed as the fold increase of each signal relative to the median fluorescence intensity of the control.). (**B**) % of cell death induced by IONP–GA/PAA (3.3 µg/mL) at 30 min, 2 h, and 6 h of treatment in the absence or presence of DFO (50 µM) and Fer1 (1 µM). Data are presented as mean ± SD from three independent experiments. *p* values were calculated with Student’s unpaired *t* test (**** *p* < 0.0001).

**Table 1 molecules-27-03970-t001:** Characterization results of the IONP–OA/TOPO and IONP–GA/PAA obtained by Vibrating Sample Magnetometry (VSM), X-ray powder diffractometry (XRD,) and Fourier-transform infrared spectroscopy (FTIR).

IONP	σ_s_ (300 K, Emu/g)	H_c_ (Oe)	DRX 2θ (°); (Diffraction Index); Phase	IR (cm^−1^); Assignment
IONP–OA/TOPO	24.7	0	35.1; (311); Fe_3_O_4_; 36.3; (111); FeO; 42.7; (200); FeO; 62.1; (220) + (440); FeO + Fe_3_O_4_	1558, 1458; $\nu_{OCO}$ (OA); 971; $\nu_{P=O}$ (TOPO);
IONP–GA/PAA	13.8	0	30.3; (220); Fe_3_O_4_; 35.7; (311); Fe_3_O_4_; 43.5; (400); Fe_3_O_4_; 57.2; (511); Fe_3_O_4_; 62.9; (440); Fe_3_O_4_.	1721; $\nu_{P=O}$ (GA) 1560, 1401; $\nu_{OCO}$ (PAA)

## Data Availability

Not applicable.

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
