# Peer review of "Novel Iron Oxide Nanoparticles Induce Ferroptosis in a Panel of Cancer Cell Lines"

_molecules, 2022, doi:10.3390/molecules27133970_

Round 1
Reviewer 1 Report
The manuscript by Gilberto L Pardo Andreu and coworkers described the biological activity of iron oxide nanoparticles against a ange of cancerous cell lines. Overall the manuscript is interesting and the hypothesised mechanism of action via the ferroptosis pathway is well supported by the experiments.
My major concern is that the title claims the use of novel nanoparticles but there is not a single line in the results section about the synthesis and formulation of these nanoparticles. The authors refer the reader to reference 23, but there is nothing in the text to tell us about the size, morphology, distribution, reproducibility... of these nanoparticles. This really need to be added to the manuscript.
Have the authors examined the sensitivity of the nanoparticles on normal cells for comparison? It would be interesting to see the selectivity.
Regarding the use of iron chelation, can the authors explain how this chelation does take place? How does DFO manage to bind to the iron atoms? Does this mean that the nanoparticles are not stable in a biological medium and therefore Fe(III) is readily available in solution? Have the authors studied the degradation of the nanoparticles over time in the relevant medium?
Author Response
Please, see the attachment.

Reviewer 2 Report
Dear Authors,
I have gone through the manuscript where authors claimed Novel Iron oxide . What kind of novality was it and how ?. expalin ?
authors discussed about induction process in results section but in methods its not clearly mentioned
Why authors did not do any phyiscal and chemcial charecterization of IONPs like Partcle size, UV , XRD, SEM etc to asses the potetaility of the formuation and in introdction of the manusscript authors claimed that done Physcochemical charecrerization .
Introduction section can be updated with explaining hypothis of the work with more refrences. Some general information quoted like 50 nanodrugs availabe so far , from the sorces 30 nanodrugs were approved from FDA and smilar sources . please correct me if im wrong .
Author Response
Please, see the attachment.

Round 2
Reviewer 1 Report
Comments from all reviewers have been taken into account. The manuscript is suitable for publication.
Reviewer 2 Report
Dear Authors,
The revised manuscript has been included physicochemical characterization and other points raised during R1 revision.